# Long Term Effects of Tillage–Crop Rotation Interaction on Soil Organic Carbon Pools and Microbial Activity on Wheat-Based System in Mediterranean Semi-Arid Region

**Sayda Jaziri** [1,2,†], **Hatem Cheikh M'hamed** [1,†], **Mohsen Rezgui** [1], **Sonia Labidi** [3], **Amir Souissi** [1], **Mounir Rezgui** [1], **Mariem Barbouchi** [1], **Mohamed Annabi** [1] **and Haithem Bahri** [4,*]

[1] Agronomic Sciences and Techniques Laboratory (LR16 INRAT 05), National Institute of Agricultural Research of Tunisia (INRAT), Carthage University, Ariana 2049, Tunisia; saydajaziri2@gmail.com (S.J.); hatemcheikh@yahoo.fr (H.C.M.); mohsenrezguig@gmail.com (M.R.); souissiamir89@gmail.com (A.S.); mounirrezgui11@gmail.com (M.R.); barbouchi.meriem@yahoo.fr (M.B.); mannabi@gmail.com (M.A.)

[2] Faculty of Sciences of Bizerte, Carthage University, Bizerte 7000, Tunisia

[3] Laboratory of Horticultural Sciences (LR13AGR01), National Institute of Agronomy of Tunisia (INAT), Carthage University, Tunis 1082, Tunisia; soniainat2002@yahoo.fr

[4] Agronomic Sciences and Techniques Laboratory (LR16 INRAT 05), National Research Institute of Rural Engineering, Water and Forests (INRGREF), Carthage University, Ariana 2049, Tunisia

\* Correspondence: haithem.bahri@ingref.ucar.tn; Tel.: +216-24010676

† These authors contributed equally to this work.

**Abstract:** Conservation agriculture based on no-tillage (NT) and crop rotation allows to enhance soil health. Based on data collected from long-term trials in a semi-arid region of Tunisia, results showed that NT increased significantly soil organic carbon stock (SOCS), soil microbial biomass carbon (SMBC), arbuscular mycorrhizal fungal (AMF) root colonization, and soil microbial respiration ($CO_2$) at 0–20 cm topsoil layer compared to conventional tillage (CT). Moreover, triennial rotation (TRI), based on annual succession of Faba bean-Durum wheat-Barley, and biennial rotation (BI), based on annual succession of Faba bean-Durum wheat, increased significantly SMBC, AMF, and $CO_2$. Likewise, a significant benefit of the two-way interactions Tillage × Rotation was observed. Furthermore, NT combined with TRI recorded the highest SOCS (2181 g C m$^{-2}$), SMBC (515 mg C kg$^{-1}$ soil), AMF (14%), and $CO_2$ which is an indicator of soil microbial respiration (1071 mg $CO_2$ kg$^{-1}$ soil). The current results highlight the benefit adoption of minimum or (NT)combined with crop diversification on soil health.

**Keywords:** tillage-rotation interaction; soil organic carbon pools; soil microbial activity; Mediterranean environment

## 1. Introduction

Soil is a tiny layer of material on the Earth's surface. It serves a critical function in regulating flow and mass and energy transmission between the lithosphere, biosphere, hydrosphere, and atmosphere [1]. It is also an important medium for human existence since it supports agricultural production, which produces 95 percent of the world's food [2]. Soils are important reservoirs of global biodiversity [1] and the second largest carbon pool on the continent after the oceans [3], since it stores approximately 1600 PgC in the first meter of soil layer [4].

Historically, soil assessments were centered on crop its production capacity, while today, worldwide community inquiries about soil health concept which is defined as the capacity of soil to function as a vital living system to sustain biological productivity, promote environmental quality, and maintain plant and animal health [5]. In fact, the use of conventional management practices based on conventional tillage causes soil degradation

as well as a loss of several soil capacities to provide ecological functions for various forms of life [6–8].

In order to improve soil quality, the authors of [4,9,10] reduced tillage systems such as minimum tillage or no-tillage (NT) with minimal soil disturbance, permanent soil surface cover with mulch of crop residues, and diverse crop rotations are among viable alternatives to conventional agriculture. The positive benefits of NT on crop production [11], economic performance [12,13], mitigating carbon dioxide emissions [14], and energy use efficiency [15] are well recognized. Indeed, NT increases soil-aggregation and its resistance to water erosion, indicating the potential for no-tilled soils to sequester organic carbon [16]. Arshad et al. [17] also showed that following NT conversion, water retention and infiltration increased due to a redistribution of pore size classes into smaller pores having the potential to improve crop water use and crop production. Moreover, soil microbial abundance and diversity are often improved under conservation tillage systems based on NT [18]. The expected benefits of NT on the soil component have contributed to the scaling of NT during the last three decades around the world, especially in the Latin American countries, USA, and Australia, and admit NT production systems as a compulsory component to sustain and enhance the global soil resource [19,20]. In addition to tillage practice, several studies identified that crop species and biomass development are the principal drivers explaining the increase of soil organic carbon, especially when crop rotations include legume species compared to cereal monoculture [21]. Legume crops are prone to reduce the use of nitrogen and phosphorus synthetic-fertilizers, to increase soil microbial activity and enzyme synthesis, as well as to stimulate some kinds of microorganisms, such as arbuscular mycorrhizal fungi, which are highly linked to the formation of soil aggregates and to the improvement of crop water uptake [22].

It is clear that the above-mentioned factors (tillage and crop species) can improve soil physical, chemical, and biological properties. In fact, the combination of these two agricultural practices could have synergistic impacts and improve soil function and services [23–25]. In addition, both NT and crop rotations have been widely adopted under conservation agriculture system [26,27], and positively affected the soil microbial biomass and activities [28,29]. Nevertheless, they have not yet been replicated in the Mediterranean area [19] such as in Tunisia where studies covering the effect of soil tillage intensity under different crop rotations are limited.

In this context based on long-term experimental trials implemented in the semi-arid region of Tunisia since 2009, this research study aims to investigate the effect of tillage combined with crop rotation on soil organic carbon pools and soil microbial activity.

## 2. Materials and Methods

### 2.1. Experimental Site

The study was conducted at the experimental station of the National Institute of Agricultural Research of Tunisia (Kef district, North West of Tunisia, 36°14′ N; 8°27′ E; 518 m). Soil samples were collected from plots of a long-term trial implemented since 2009 (before the field was uncultivated). The experimental site is located in the Boulifa region (Figure 1). An initial soil characterization was done, the soil is a silty clay belonging to Entisols with 51% of clay, 30% of silt, 19% of sand and a pH ($H_2O$, 1:2) of 8.2 [30] and the initial stock of soil organic carbon (SOCS) is about 1486 g C m$^{-2}$.

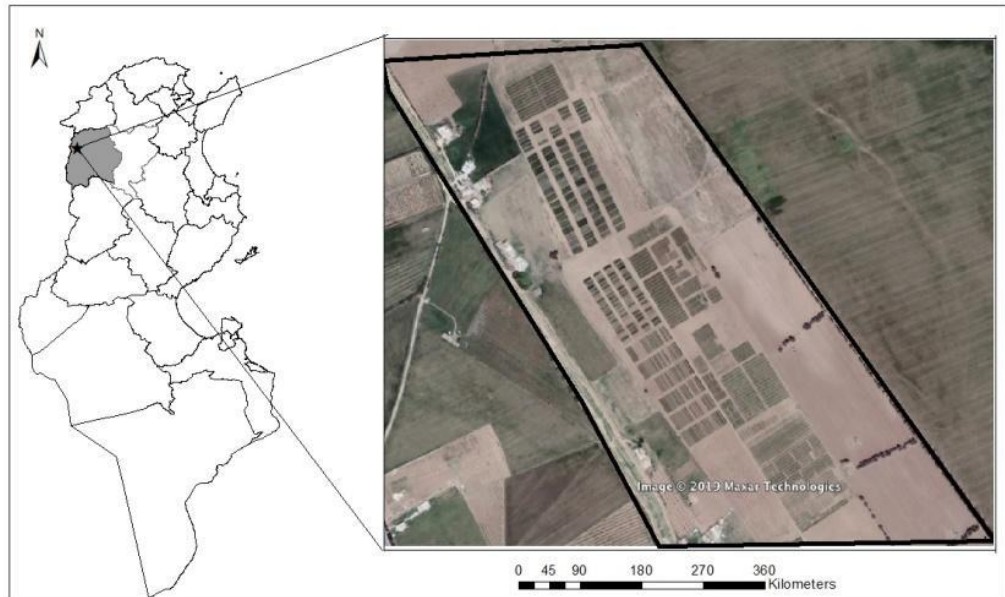

**Figure 1.** Location of the experimental site at the INRAT–Agricultural Experimentation Unit of kef.

The trial was implemented according to a split-plot design. Four replications and two factors were considered: the tillage practices (Tillage) as the main factor and the crop rotation (Rotation) as a sub-factor. Three treatments of tillage practice are tested: (i) Conventional Tillage (CT): inversion of the first 30 cm depth of soil with two plow coulters and a mouldboard, followed by clod fragmentation with an offset sprayer; (ii) Moderate Tillage (MT): tillage without soil turning using a chisel with rigid tine followed by a Canadian cultivator with vibrating tines; and (iii) No-till (NT): using NT seeder after the application of glyphosate N-(phosphonomethyl)-glycine. For CT and MT plots, sowing was performed using a conventional tine seeder. Three crop rotation sequences are tested: (i) Durum wheat monocropping (M); (ii) biennial rotation (BI) based on annual succession of faba bean and durum wheat; and (iii) triennial rotation (TRI) based on annual succession of faba bean, durum wheat, and barley. Each phase of the tested rotations was presented in each year.

### 2.2. Weather Conditions Monitoring

Weather data were recorded daily during the period 2009–2019. The average annual rainfall (from September to August), from 2009 to 2018, at the Kef site was 450 mm. Meanwhile, the 2018–2019 growing season was considered as a wet season (691 mm) (Figure 2). Monthly maximum air temperature (Tmax) varied between 16.2 °C in January 2019 and 43.5 °C in June 2019. Monthly minimum air temperature (Tmin) varied between $-1.3$ °C in January 2019 and 14.6 °C in August 2019. On the other hand, June, July, and August 2019 were hotter than that of the long-term average (43.5 vs. 39.6, 43 vs. 40.8, and 43.2 vs. 41.2 °C for Tmax, respectively).

### 2.3. Soil Sampling

Soil Samples were collected in February 2019, from the topsoil (0–20 cm soil layer) with three replicates of durum wheat sub-plots. The soil samples were homogenized and sieved at 2 mm after removing the particulate plant material and then divided into three sub-samples. The first sub-samples were oven-dried at 105 °C to estimate the soil moisture at each sampling date. The second sub-samples were air-dried before being used for the measurement of soil organic carbon content and microbial respiration, and the third sub-samples were directly stored at 4 °C before being used for the assessment of the microbial biomass size. Furthermore, soil bulk density in each treatment was measured on 5-cm diameter undisturbed soil cores for the 0–20 cm soil depth.

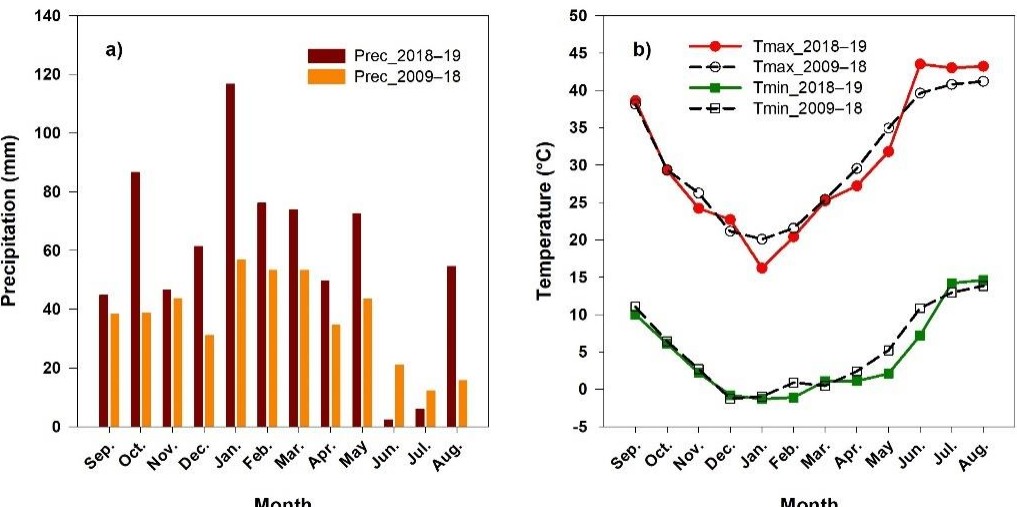

**Figure 2.** (**a**) Monthly precipitation (Prec), (**b**) minimum air temperature (Tmin), and maximum air temperature (Tmax) during the growing season (2018–2019) and their long-term averages (2009–2018) at experimental station.

### 2.4. Soil Analysis

#### 2.4.1. Soil Organic Carbon Stock

Soil organic carbon (SOC) was determined by wet oxidation method [31]. Briefly, 0.2 g of air-dried 2 mm sieved soil was oxidized with 5 mL of bichromate of potassium ($K_2Cr_2O_7$; 1 N) in presence of 10 mL of sulfuric acid ($H_2SO_4$; 18 M). The SOC content was determined by spectrophotometric at 660 nm using a calibration curve prepared with different concentrations of glucose in solution. Soil organic carbon stock (SOCS) was computed as the product of SOCcontent and bulk density.

#### 2.4.2. Soil Microbial Biomass Carbon

Soil microbial biomass carbon (SMBC) was estimated using the fumigation–incubation method [32]. For each treatment, six portions (25 g) of the moist soil stored at 4 °C were weighed into small glass vials. Three portions were fumigated in a steel vacuum chamber (Labconco®, Kansas City, MI, USA) for 24 h in the presence of free-ethanol chloroform ($CHCl_3$), and three portions were unfumigated. The fumigated and unfumigated portions were then placed in separate jars and were incubated during 10 and 20 days respectively. The incubation was carried on in darkness at 28 °C. The released $CO_2$ was trapped in 10 mL of NaOH (1 N), replaced after 2, 5, 10, and 20 days, and analyzed by acid titration (HCl; 0.1 M) in the presence of $BaCl_2$. SMBC was calculated using the Equation (1).

$$SMBC \left(mg\,C\,kg^{-1}\right) = \frac{(X - y)}{k} \tag{1}$$

where X: the $CO_2$ evolved by fumigated soil over the period 0–10 days; y: the $CO_2$ evolved by unfumigated soil over the 10–20-day period; k: the conversion factor used to shift chloroform labile C to microbial biomass C, it was set at 0.5 according to Jenkinson and Powlson [32].

#### 2.4.3. Soil Microbial Respiration

Soil microbial respiration ($CO_2$) was determined under controlled laboratory conditions by the respirometry test method [33]. For each treatment, 25 g of air-dried soil were incubated in hermetically sealed jars. The soil samples were wetted to their water-holding capacity by adding deionized water. The incubation was carried during 90 days in darkness at 28 °C. In each glass jar, organic carbon that was mineralized to carbon dioxide ($CO_2$) was trapped in 10 mL of sodium hydroxide (NaOH; 1 N). These traps were periodically replaced after 2, 4, 7, 10, 14, 20, 32, 45, 60, and 90 days of incubation and the $CO_2$ released

was dosed using titrimetric method with hydrochloric acid (HCl; 0.1 M) in presence of barium chloride ($BaCl_2$).

### 2.4.4. Arbuscular Mycorrhizal Fungal Root Colonization

The estimation of the arbuscular mycorrhizal fungal (AMF) root colonization was done on the fresh roots of wheat as described by Phillips and Hayman [34]. Three plants per plot for each treatment were sampled in early May 2019 at the grain filling stage of wheat. Wheat roots were cut and then soaked in a hydrogen peroxide ($H_2O_2$, 30%) bath for 5 min to remove pigments before being cleared in potassium hydroxide (KOH, 10%) and stained with Trypan blue (0.05%). Root mycorrhizal rates were calculated using the method described by McGonigle et al. [35]. For each plant, microscopic slides with 15 randomly selected root fragments (1 cm) were observed under an optical microscope with a magnification of 100. The arbuscular mycorrhizal fungal (AMF) root colonization was calculated using Equation (2).

$$AMF \ (\%) = \frac{(\,G - p\,)}{G} \times 100 \tag{2}$$

where: G: Total number of observed intersections; p: no observed mycorrhizal structure.

### 2.5. Statistical Analysis

For each dependent variable, significant differences between treatments were performed by two-way ANOVA to explore the effect of tillage and rotation and their interaction. In order to compare the differences among treatment means ($p < 0.05$), Fisher's least significant difference (LSD) was performed using STATISTIX 7.0 (Analytical Software, Tallahassee, FL, USA) [36]. Simple linear regression was used to detect the relationship between variables studied. Plots were drawn with R3.4.2 software (R Foundation for Statistical Computing, Vienna, Austria) [37].

### 3. Results

ANOVA showed that tillage practices (Tillage), crop rotation (Rotation) and their interaction (Tillage × Rotation) affected significantly, ($p < 0.05$ to $p < 0.001$), the following soil properties: soil organic carbon stock (SOCS), soil microbial respiration ($CO_2$), soil microbial biomass carbon (SMBC), and arbuscular mycorrhizal fungal (AMF) root colonization (Table 1).

**Table 1.** Significance from ANOVA testing Tillage effect (No-till: NT, Moderate tillage: MT, and Conventional tillage: CT) and Rotation effect (Monocropping: M, biennial: BI, and triennial: TRI) and their interaction (Tillage × Rotation) effect on soil organic carbon stock (SOCS), soil microbial respiration ($CO_2$), soil microbial biomass carbon (SMBC), and arbuscular mycorrhizal fungal (AMF) root colonization. Data are averages ± standard errors. Different lowercase letters indicate significant differences between all treatments in each item ($p < 0.05$).

| Source of Variation | SOCS (g C m$^{-2}$) | CO$_2$ (mg CO$_2$ kg$^{-1}$) | SMBC (mg C kg$^{-1}$) | AMF (%) |
|---|---|---|---|---|
| Tillage | *** | ** | *** | *** |
| NT | 2146.0 ± 30.4 [a] | 1004.8 ± 20.2 [a] | 402.8 ± 10.7 [a] | 13.4 ± 0.2 [a] |
| MT | 1951.2 ± 30.4 [b] | 943.3 ± 20.2 [b] | 313.3 ± 10.7 [c] | 12.6 ± 0.2 [b] |
| CT | 1856.8 ± 30.4 [c] | 926.1 ± 20.2 [b] | 342.3 ± 10.7 [b] | 12.4 ± 0.2 [b] |
| Rotation | NS | *** | *** | *** |
| M | 1942.5 ± 30.4 [a] | 936.6 ± 20.2 [b] | 333.3 ± 10.7 [b] | 12.2 ± 0.2 [b] |
| BI | 2016.9 ± 30.4 [a] | 918.6 ± 20.2 [b] | 299.6 ± 10.7 [c] | 13.2 ± 0.2 [a] |
| TRI | 1994.6 ± 30.4 [a] | 1019.0 ± 20.2 [a] | 425.5 ± 10.7 [a] | 13.0 ± 0.2 [a] |
| Tillage × Rotation | ** | * | *** | *** |

*** < 0.001, ** < 0.01, * < 0.05, NS ≥ 0.05.

### 3.1. Carbon Pools

#### 3.1.1. Soil Organic Carbon Stock

Compared to the initial soil organic carbon stock (SOCS) in 2009 (1486 g C m$^{-2}$) a significant improvement of SOCS in the 0–20 cm layer was observed in all treatments. The highest increase of SOCS between 2009 and 2019 was observed in the NT-TRI treatment (695 g C m$^{-2}$) followed by the NT-BI treatments (672 g C m$^{-2}$) while the lowest increase was obtained in the CT-BI (318 g C m$^{-2}$) and the CT-M (342 g C m$^{-2}$) treatments.

The statistical analysis showed that tillage practices (Tillage) affected significantly SOCS (Table 1). The SOCS on the 0–20 cm soil layer averaged across treatments was significantly higher under NT (2146 g C m$^{-2}$) than under MT (1951 g C m$^{-2}$), which in turn was significantly higher than under CT (1856 g C m$^{-2}$). Nevertheless, crop rotation (Rotation) did not significantly affect SOCS (Table 1). However, crop rotation also significantly affected SOCS when it is combined with tillage practices (Figure 3). NT combined with biennial or triennial rotation recorded a significant increase of SOCS. Indeed, the NT-TRI and NT-BI treatments had the highest SOCS, with respectively 2181 g C m$^{-2}$ and 2158 g C m$^{-2}$. However, CT treatment combined with monocropping recorded the lowest SOCS, which ranged from 1803 g C m$^{-2}$ and 1934 g C m$^{-2}$ (Figure 3).

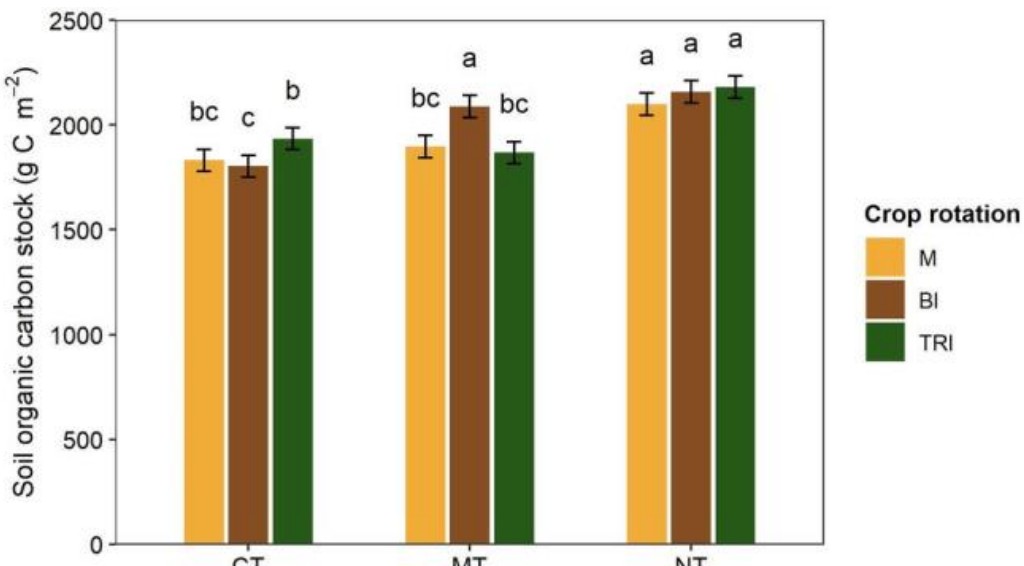

**Figure 3.** Effect of two-way interactions of 'Tillage × Rotation' on the soil organic carbon stock (SOCS, g C m$^{-2}$). Tillage effect (No till: NT, moderate tillage: MT, and conventional tillage: CT) and Rotation effect (Monocropping: M, biennial: BI, and triennial: TRI). Data are averages ± standard errors. Different lowercase letters indicate significant differences between treatments ($p < 0.05$).

#### 3.1.2. Soil Microbial Biomass Carbon

Soil microbial biomass carbon (SMBC) was significantly higher under NT (401 mg C kg$^{-1}$ soil) than under CT (363 mg C kg$^{-1}$ soil), which in turn was significantly higher than under MT (297 mg C kg$^{-1}$ soil). On the other side, crop rotation showed a significant effect on SMBC ($p < 0.05$). Likewise, TRI recorded the highest size of SMBC (424 mg C kg$^{-1}$ soil). Wheat monocropping and biennial rotation had statistically similar SMBC levels (325 and 312 mg C kg$^{-1}$ soil, respectively). Tillage × Rotation significantly affected ($p < 0.001$) SMBC size, with higher levels observed under NT-TRI (515 mg C kg$^{-1}$ soil) compared to the other treatment combinations (Figure 4).

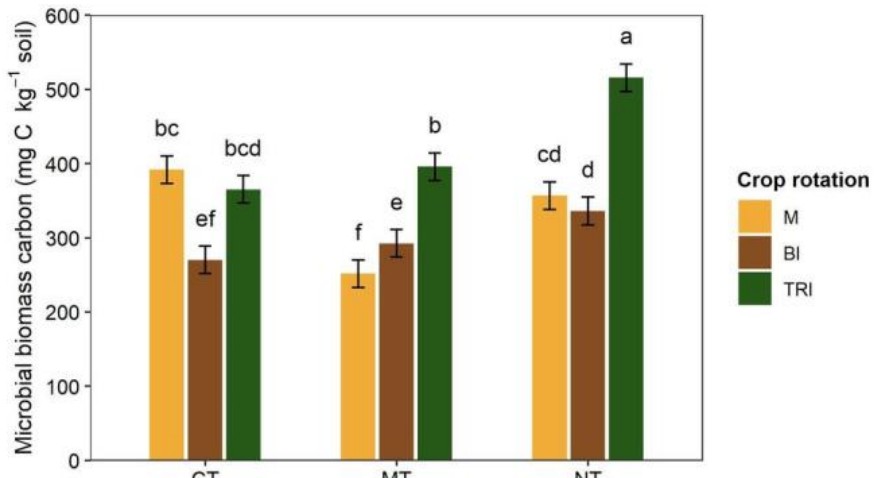

**Figure 4.** Effect of two-way interactions 'Tillage × Rotation' on soil microbial biomass carbon (SMBC, mg C kg$^{-1}$ soil). Tillage effect (No till: NT, moderate tillage: MT, and conventional tillage: CT) and Rotation effect (Monocropping: M, biennial: BI, and triennial: TRI). Data are averages ± standard errors. Different lowercase letters indicate significant differences between treatments ($p < 0.05$).

### *3.2. Soil Microbial Activity*

### 3.2.1. Soil Microbial Respiration

The cumulative released $CO_2$ by soil microbial respiration after 90 incubation days under different treatments is given in Figure 5. The $CO_2$ released was significantly affected by tillage practices and crop rotation. The $CO_2$ was significantly higher under NT (1005 mg $CO_2$ kg$^{-1}$ soil) than under MT (943 mg $CO_2$ kg$^{-1}$ soil) and under CT (926 mg $CO_2$ kg$^{-1}$ soil). MT and CT had a statistically similar cumulative $CO_2$ pattern. On the other side, the $CO_2$ released after 90-day incubation was significantly higher with triennial rotation (1019 mg $CO_2$ kg$^{-1}$ soil) compared to the wheat monocropping (936 mg $CO_2$ kg$^{-1}$ soil) and to the biennial rotation (919 mg $CO_2$ kg$^{-1}$ soil), which were statistically similar. Moreover, results showed that the interaction of Tillage × Rotation has significantly affected $CO_2$. NT-TRI (1071 mg $CO_2$ kg$^{-1}$ soil) and MT-TRI (1052 mg $CO_2$ kg$^{-1}$ soil) released the most $CO_2$ during 90-day incubation (Figure 5).

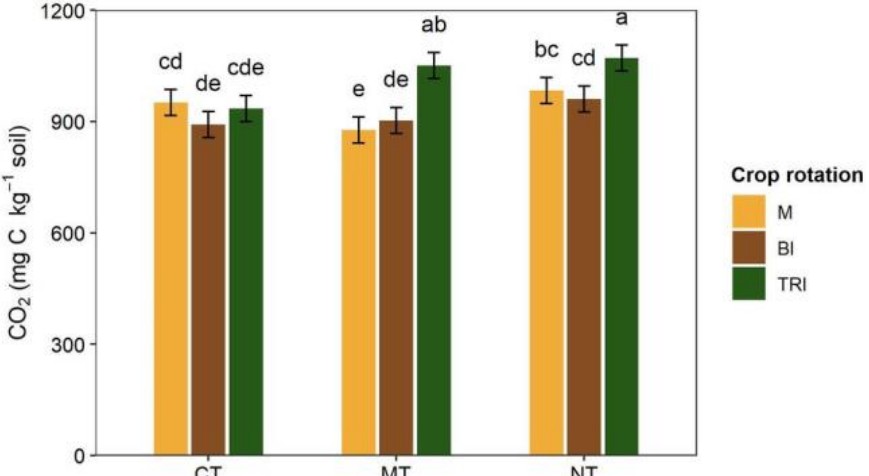

**Figure 5.** Effect of two-way interactions 'Tillage × Rotation' on total $CO_2$ released at 90-day incubation. Tillage effect (No till: NT, moderate tillage: MT, and conventional tillage: CT) and Rotation effect (Monocropping: M, biennial: BI, and triennial: TRI). Data are averages ± standard errors. Different lowercase letters indicate significant differences between treatments ($p < 0.05$).

### 3.2.2. Arbuscular Mycorrhizal Fungal Root Colonization

The arbuscular mycorrhizal fungal (AMF) root colonization at the wheat grain filling stage ranged between 10% and 14% (Figure 6). The AMF was significantly higher under NT (13.5%) compared to MT (12.7%) and CT (12.3%). MT and CT had statistically similar AMF rates. AMF rate was significantly higher in wheat-roots sampled from biennial rotation (13.2%) and triennial rotation (13.1%) plots than those from the wheat-monocropping plots (12.2%). On the other hand, crop rotation significantly affected mycorrhizal root colonization rate when it is combined with tillage practices. In fact, the NT-TRI and MT-TRI had the highest AMF rates (14%).

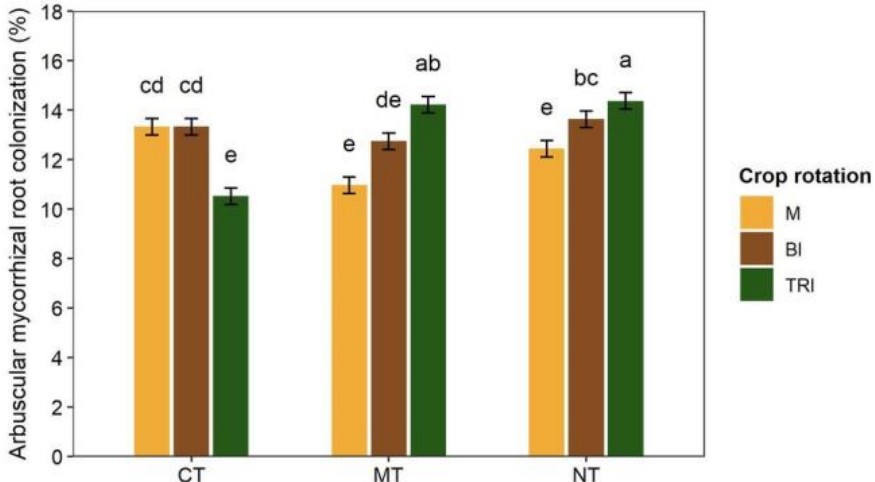

**Figure 6.** Effect of two-way interactions 'Tillage × Rotation' on arbuscular mycorrhizal root colonization (%). Tillage effect (No till: NT, moderate tillage: MT, and conventional tillage: CT) and Rotation effect (Monocropping: M, biennial: BI, and triennial: TRI). Data are averages ± standard errors. Different lowercase letters indicate significant differences between all treatments in each item ($p < 0.05$).

### 3.3. Relationship between Soil Carbon Pools and Soil Microbial Activity Variables

Figure 7 shows the correlation matrix chart of soil carbon pools and soil microbial activity variables under different tillage practices and crop rotation. SMBC showed a significantly positive correlation with $CO_2$ ($r = 0.78$). AMF was considerably correlated with $CO_2$ ($r = 0.55$) and SMBC ($r = 0.47$). However, SOCS showed the lowest correlation with other parameters studied ($CO_2$, $r = 0.35$; SMBC, $r = 0.36$; AMF, $r = 0.22$).

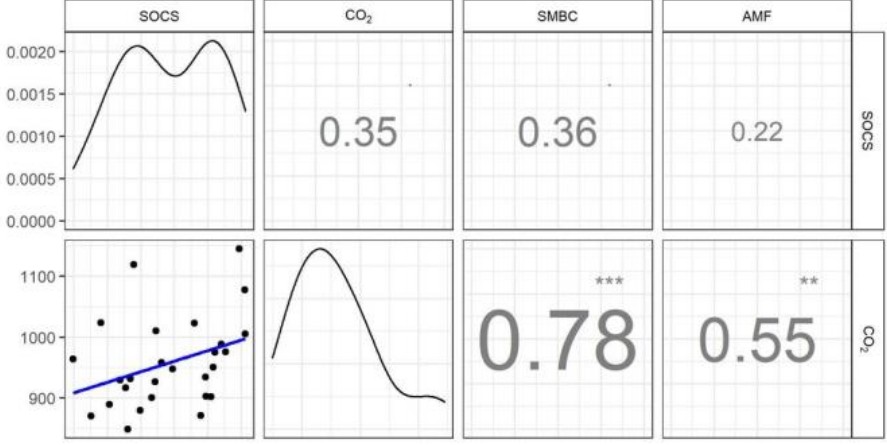

**Figure 7.** *Cont.*

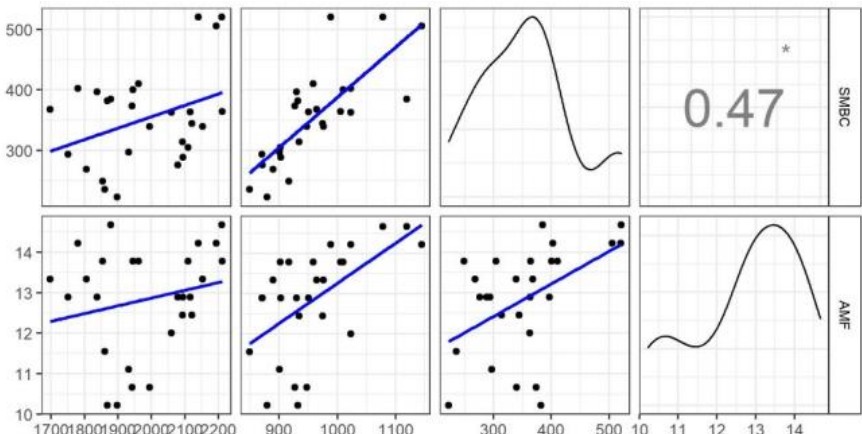

**Figure 7.** Correlation matrix chart of chemical and biological variables under different tillage practices and crop rotation. Soil organic carbon stock (SOCS), soil microbial respiration ($CO_2$), soil microbial biomass carbon (SMBC), and arbuscular mycorrhizal fungal (AMF) root colonization. The upper panels show Pearson's correlation coefficients while the lower panels report the scatter plots. \*, \*\*, \*\*\* indicates significance at $p < 0.05$, $< 0.01$, and $0.001$, respectively.

## 4. Discussion

Soil organic carbon is a crucial factor for soil quality and crop productivity [38,39]. Compared to the initial level, the cultivation of the field induced an increase of the soil organic carbon stock (SOCS) estimated to +25%, +31%, and +44% for conventional tillage (CT), moderate tillage (MT), and no-till (NT) treatments, respectively. Nevertheless, its stock in the soil arable layer declines over time with intensive agriculture based on energetic ploughing and limited crop rotation adoption [40,41]. In this context, the present study showed that, over 10-year management, conservative tillage options had a significant influence on SOCS in the 0–20 cm soil layer with an increase of 15% for NT and 10% for MT treatments compared to CT. However, according to a previous study in Tunisia, Ben Moussa-Machraoui et al. [42] did not observe a significant increase in soil organic carbon (SOC) after four years of NT adoption against CT ones. For a 36-year long-term experiment in Canada, Laamrani et al. [43] demonstrated that SOC was significantly higher for NT (2.73%) compared to CT (2.51%). Moreover, some long-term studies in eastern Canada found that NT produced more carbon in topsoil than CT [44,45]. In the Mediterranean conditions, characterized by the low quantity of residue retention after harvesting, the increase of SOC under NT is due mainly to the duration of the conversion from CT to NT. In fact, Dimassi et al. [46] and Follett [47] explained the SOC increase under NT by the increase of SOC inputs and a decrease of SOC decomposition, or a combination thereof. The meta-analysis done by Virto et al. [48] showed that the variation in SOC input was a main factor explaining the variability in SOC storage due to tillage. The total input of SOC is highly related to cropping frequency and crop biomass returned to the soil, including below-ground materials. Several studies conducted in different regions and under different cropping systems, showed an increase in SOC under NT with residue retention (ranging from 0.22 to 1.80 t ha$^{-1}$ year$^{-1}$) compared to CT [49,50]. Adopting a modelling approach, Bahri et al. [51] showed, in Tunisia, an increase of SOC accumulation ranged from 0.12 to 0.31 t ha$^{-1}$ year$^{-1}$ between NT with residue retention compared to NT without residue retention.

The SOCS gain is enhanced when NT is combined with biennial (BI) and triennial (TRI) rotations. Under NT-TRI treatment, SOCS increased by 19% compared to CT-M. In fact, several studies showed that cereal–legume systems are effective for increasing soil organic status [52,53]. Indeed, legume crops can provide both organic matter and nitrogen to the soil [52], and thus enhance soil fertility. Carbon storage in the soil by converting from CT to NT can also conserve nitrogen in soil because SOC and total nitrogen levels are highly correlated [54]. According to a meta-analysis study, Luo et al. [55] showed



over a mean duration of 11 years, an increase of 5% of SOC when it was combined to a monoculture and an increase of 18% when perennial legumes were introduced in the rotation. Results recorded in the framework of this study under a semi-arid environment, showed a substantial SOCS gain of 3% and 4%, respectively, under TRIand BIrotation treatments compared to monocropping. The combination of NT and crop rotation has the potential to increase SOCand nitrogen further as a result of higher biomass yields due to reduced soil evaporation and improved water use efficiency, particularly in semi-arid areas [56,57]. Likewise, the positive effects of crop rotations on physical, chemical, and biological soil properties could be attributed to higher SOC inputs and the diversity of plant residues returned to soils [58,59]. These results confirm the importance of crop diversification, as a pillar, to success conservation agriculture adoption in semi-arid regions.

The adoption of NT combined with BIand TRIcrop rotations significantly improved the soil microbial activity. In fact, compared to CT with full inversion, NT is supposed to decrease SOC mineralization rates. This is due to less favourable local climatic conditions and better physical protection of organic matter in soil aggregates [60]. Thus, long-term adoption of NT is considered as a relevant agronomic option to increase SOC sequestration, which contributes to the reduction of $CO_2$ released into the atmosphere [61]. However, several studies that focused on the superficial soil layers found higher mineralization rates of carbon under NT that could be attributed to the relative increase of soil organic matter and particulate organic matter content in the surface layers under NT [4,9,10]. Numerous studies found that the concentration of organic carbon in surface soil and subsoil was similar or slightly greater under non-tilled soil than tilled one [29,62,63]. Balesdent et al. [60] and Paustian et al. [64] estimated the average of SOC residence time under NT was 1 to 2.1 times higher than under full inversion tillage.

The tillage systems significantly affected soil microbial biomass carbon (SMBC) size and respiration. Indeed, soil microbial respiration ($CO_2$) and microbial biomass were higher under NT, which is a result of higher SOC content recorded under NT. Several findings reported an increase in soil microbial level and respiration under the NT system [65,66]. In fact, the reduced soil disturbance under the minimum tillage system provides a steady source of organic carbon that consequently improves microbial activity and various factors, such as higher moisture content and higher soil aggregation [16,67]. Although a high SOCS is an important factor contributing to SMBC content [68]. The current study found a low correlation ($r = 0.35$) between these two parameters. Besides, the inclusion of legumes in wheat-based cropping systems had a significant effect on overall soil chemical and biological variables, which are the key soil variables liable to improve soil quality. Our results are in agreement with those reported by Qin et al. [69] observed an improvement of soil microbial communities and enzymes for cropping systems based on different crop rotations with the introduction of legumes compared to monoculture. In our case, SMBC size ($r = 0.78$) and AMF ($r = 0.55$) are both strongly linked with microbial respiration ($CO_2$), a worldwide indicator of heterotrophic microorganisms.

Despite its small size, the SMBC pool is an important labile fraction of SOC and constitutes a good indicator of early changes in cropland management practices that impact soil health and agricultural production [70,71]. Soil microbial biomass plays a significant role in enhancing soil aggregation, promoting carbon and nitrogen turnover, and thus nutrient cycling [71,72]. Tillage and crop rotation induce changes in soil physical and chemical properties, and consequently, in soil microbial dynamics [73]. In the present study, the soil microbial biomass was affected by tillage practices, crop rotation, and by their interaction. In fact, an increase of SMBC about 17% under NT compared to CT was observed. Regarding tillage effect, many authors reported greater microbial biomass under NT due to more favourable microclimates in comparison with CT [74,75]. Crop species diversity, through diversification of root growth, distribution in the soil, and biomass retention in the soil after harvesting, contributes also to soil microbial richness, and root growth [55]. The SMBC gain in TRI (+42%) and BI (+12%) treatments compared to monoculture indicated that SMBC was highly responsive to tillage suppression [76].

Furthermore, our results demonstrated that reduced tillage and crop rotation improved the arbuscular mycorrhizal fungal (AMF) root colonization by +8.7% under NT and +7.9% in rotation compared to CT and monoculture, respectively. Results obtained are in agreement with those of Roldán et al. [77], who observed the highest levels of mycorrhizal propagules in the no-till soil compared to tillage soils. Taibi et al. [78] observed an increase of durum wheat mycorrhization rate after 3-year adoption of NT in Algeria. NT practice causes less frequent soil disturbance and fungal hyphae have a greater opportunity to be developed with surface residue retention, thus promoting fungal abundance and diversity [79,80]. Therefore, total mycorrhizal root colonization plays a critical role in the turnover of C and N through their hyphal networks [81] and offers an additional power of water uptake by crop roots [82].

## 5. Conclusions

Based on a long-term experimentation of wheat-based system, this study shows that the minimum tillage or no tillage combined with crop diversification (two pillars of conservation agriculture) under semi-arid conditions allows enhancing soil carbon pools and soil microbial activity which are the main indicators of the soil health. Indeed, MT and NT enhanced soil organic carbon stock by + 5% and +16%, respectively. The same trend was observed for soil microbial biomass carbon with an increase about +18% for NT compared to CT. Moreover, Crop rotation and mainly triennial rotation has positively affected soil microbial respiration (+9%), soil microbial biomass carbon (+28%), and arbuscular mycorrhizal fungal root colonization (+7%) compared to wheat monocropping. This result shows clearly the positive effect of crop diversification on soil microbial activity. An additional beneficial effect of crop rotation on soil health when it is combined with NT or MT. The current results are promising and helping the adoption of conservative practices in Tunisia where more than 50% of cultivated soils are threatened by water erosion in the semi-arid regions of the Mediterranean basin.

**Author Contributions:** Conceived and designed the experiments, M.R. (Mohsen Rezgui), H.C.M. and H.B.; performed the experiments, S.J., M.R. (Mohsen Rezgui), H.C.M. and M.R. (Mounir Rezgui); analyzed the data, S.J., A.S., M.R. (Mohsen Rezgui), S.L. and M.B.; wrote the paper, S.J., H.C.M. and H.B.; assisted with writing, reviewing, and editing the paper, M.A., H.B. and H.C.M.; supervised the work, H.B., M.R. (Mohsen Rezgui) and M.A. All authors have read and agreed to the published version of the manuscript.

**Funding:** This study was supported by the Agronomic Sciences and Techniques Laboratory of the National Institute of Agricultural Research of Tunisia (INRAT), Carthage University, the CLCA-II project and the CAMA project. CLCA project Phase II: 'Use of conservation agriculture in crop–livestock systems in the drylands for enhanced water use efficiency, soil fertility and productivity in NENA and LAC countries' funded by the International Fund for Agricultural Development (IFAD) (ICARDA's agreement N_200116). CAMA project: 'Research-based participatory approaches for adopting Conservation Agriculture in the Mediterranean Area', funded by PRIMA Foundation (Call 2019 Section 1, grant agreement N_1912).

**Data Availability Statement:** Not applicable.

**Acknowledgments:** Thanks to the team of the Agronomic Sciences and Techniques Laboratory of the National Institute of Agricultural Research of Tunisia (INRAT), Carthage University for help and support.

**Conflicts of Interest:** The authors declare no conflict of interest.

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
