# Peer review of "Long Term Effects of Tillage–Crop Rotation Interaction on Soil Organic Carbon Pools and Microbial Activity on Wheat-Based System in Mediterranean Semi-Arid Region"

_agronomy, doi:10.3390/agronomy12040953_

Round 1
Reviewer 1 Report
The reviewed manuscript describes the long-term effects of different tillage practices and crop rotation on soil health. The manuscript seems exciting but needs some important explanations before final publications.
- First of all, the Soil sampling period information is unclear. Authors describe the long-term effect on tillage systems based on only one soil sampling period (February 2019)?! In this context, also the aim of this work and conclusions doesn't match.
- Before the 2009 trial, what were the differences in analyzed parameters between the tillage systems? I suggest comparing the years 2009 and 2019 of the study experiment.
- All analyzed in this work soil parameters are very dependent on meteorological conditions. Therefore, please provide more information about meteorological conditions during 2019, e.g., temperature and precipitation?
- All obtained results should be present during the particular months of the vegetation period 2019 or as the mean from this months and then compared with the start of the experiment - 2009?!
- There is no information about soil sampling replications. How many repetitions include in the statistical analysis?
- Line 95- pH in which solution? Requires more detail.
- Line 264-266 - the citation indicated.
Recommendation - major revision
Author Response
Response to Reviewer 1 Comments
Point 1: English language and style are fine/minor spell check required
Response 1: The English language of the manuscript was revised.
Point 2: Introduction section must be improved
Response 2: The introduction section is reorganised and amended.
Point 3: Research design must be improved
Response 3: To improve the research design:
- The objective was reformulated to emphasis expected scientific messages.
- The material and methods section was reorganised and some methods and protocols were clarified.
- The results section was improved and reorganised, in fact authors merged the soil organic carbon stock and microbial biomass under a sub section entitled “carbon pools” and the soil respiration and the Arbuscular mycorrhizal fungal under a sub section entitled “soil microbial activity”.
- The conclusion was revised and was rewritten
These modifications improved the state of the art of the study, the methodology and the presented results.
Point 4: Results presentation must be improved
Response 4: Results presentation was improved through i) a change of figures (format, histograms colour, organisation), ii) According to the next comment (Point 10), the authors reorganised the results into parts. The first one is “carbon pools” and the second one “soil microbial activity”. This organisation will improve the logical flow of the scientific messages of the present paper.
Point 5: Conclusion supported by results must be improved
Response 5: The conclusion was revised and reorganised.
Point 6: The Soil sampling period information is unclear. Authors describe the long-term effect on tillage systems based on only one soil sampling period (February 2019)?! In this context, also the aim of this work and conclusions doesn't match.
Response 6: As mentioned in section 2.1 “The trial was implemented in 2009” in the experimental station of the National Institute of Agricultural Research of Tunisia and the soil samples were collected in February 2019. In this sense, obtained results correspond to the long-term effect (10 years) of the following treatments. The comparison was done in relative, the monocropping under tilled condition is the control (business as usual). However, at the beginning of the trial implementation, the classical soil characterisation was done, including texture, pH and soil organic matter content. So, we added in the manuscript, a paragraph mentioning the evolution of the TOC between 2009 and 2019 for the different treatments.
Point 7: Before the 2009 trial, what were the differences in analyzed parameters between the tillage systems? I suggest comparing the years 2009 and 2019 of the study experiment.
Response 7: The comparison was done in relative, the monocropping under tilled condition is the control (business as usual). However, at the beginning of the trial implementation, the classical soil characterisation was done, including texture, pH and soil organic matter content. So based on the initial level of soil TOC, we added in the manuscript: the initial soil carbon stock to the “experimental site “subsection and a paragraph mentioning the evolution of the TOC between 2009 and 2019 for the different treatments was added to the “soil organic carbon stock” subsection.
The soil cultivation, mainly using sustainable practices, led to the increase of SOCS compared to wheat-fallow option previously followed on the plot where the trial had been set up.
Point 8: All analyzed in this work soil parameters are very dependent on meteorological conditions. Therefore, please provide more information about meteorological conditions during 2019, e.g., temperature and precipitation?
Response 8: A new subsection “Weather conditions monitoring” was added to describe the climate (annual rainfall, Tmin and Tmax) of the experimental site during 2018-2019 season and the long-term average (2009-2018)
Point 9: All obtained results should be present during the particular months of the vegetation period 2019 or as the mean from this months and then compared with the start of the experiment - 2009?!
Response 9: We did a relative comparison. The monocropping under tilled condition is the control (business as usual). In 2009 only a classical soil characterisation was done but not the soil microbial parameters. So, we added in the document the evolution of SOCS from the initial stage to 2019.
Point 10: There is no information about soil sampling replications. How many repetitions include in the statistical analysis?
Response 10: Soil sampling was carried out with three replications for each plot. This information was added to the “soil sampling” subsection.
For statistical analysis, we considered 4 replications of plots x 3 replications of soil samples
Point 11: Line 95- pH in which solution? Requires more detail.
Response 11: Soil pH was measured using a digital pH meter (AD1000, Adwa Instruments) after extraction in deionized water (soil:water = 1:2). This information was added to the “experimental site” subsection. (L93)
Point 12: Line 264-266 - the citation indicated
Response 12: Different references were added for the paragraph: Reeves, 1997; Lal, 2020; Drinkwater and Snapp, 2007 and Kay and VandenBygaart, 2002. (L285-289)
Reviewer 2 Report
Dear Authors.
This study is relevant, considering it is related with a long-term experiment in which NT x rotation treatments were tested in plots installed (since 2009) in the Mediterranean Region (Tunisia). The manuscript is well written, though the main effects of NT and crop rotation on the variables analyzed were not adequately revised: the introduction section must be improved. Soil sampling is an important issue in this kind of study. This is the reason why soil layers collected should be stratified taking into account the treatments tested: thus, it is right to collect samples in tinny soil layers (e.g. 0-5 cm) of the NT plots, but it is not advisable to collect samples in stratified way in layers where the soil was mixed up to 30 cm.... Thus, it is advisabl to show the dataset following the principles of soil sampling and the effects of treatments normally reported for NT and CT plots. The influence of crop rotation on soil C pools and soil biota activity is related with the amount and composition of biomass added to soil in each agricultural season. Thus, to have a correct nexus of causality, it would be advisable to present in a table all the data related to crop yield or cover crop biomass production for each agricultural season. Soil Health (“the capacity of soil to function as a vital living system, within ecosystem and land-use boundaries, to sustain plant and animal productivity, maintain or enhance water and air quality, and promote plant and animal health” (Doran and Zeiss, 2000)) is an important and ample concept that could not be used in this context. It was only evaluated 2 soil C pools and microbial activity and population of only a mycorrhiza group... Wheat as the main cash crop should be mentioned in the title of the manuscript, as well as the terms "C pools", "tillage-rotation interaction" and "microbial activity in a clayey Mediterranean soil... Increased activity of soil biota (microbial respiration, CO2) is common in NT areas due to a higher labile C content, increased stocks of C in soil surface, more substrate for microorganisms etc... Thus, increased stock of C in NT soils and enhanced activity of soil biota in the same plots are not conflicting issues in this kind of experiment.. Mycorrhiza is not only the driving factor for a higher soil biota respiration in NT areas... Considering the points raised in this report, a new full revised version of the manuscript is required for further analysis of the real scientific merit of the manuscript.
Author Response
Response to Reviewer 2 Comments
Point 1: English language and style; Moderate English changes required
Response 1: The English language of the manuscript was revised.
Point 2: Introduction must be improved
Response 2: The introduction section is reorganised and amended. According to the comment linked to the soil health concept (Point 10), a paragraph was added in order to introduce this concept for the following sections (Results and discussion).
The objective of the study was written according to the comments given in Point 5.
Point 3: Research design can be improved
Response 3: To improve the research design:
- The objective was reformulated to emphasis expected scientific messages.
- The material and methods section was reorganised and some methods and protocols were clarified.
- The results section was improved and reorganised, in fact authors merged the soil organic carbon stock and microbial biomass under a sub section entitled “carbon pools” and the soil respiration and the arbuscular mycorrhizal fungal under a sub section entitled “soil microbial activity”.
- The conclusion was revised and was rewritten
These modifications improved the state of the art of the study, the methodology and the presented results.
Point 4: Methods description must be improved
Response 4: The method description was reorganised to follow the same plan than results description section. The section linked to the arbuscular mycorrhizal fungal root had improved to simplify the explanation of the used method and a reference was added according to the used method (McGonigle T.P., Miller M.H., Evans D.G., Fairchild G.L. and Swan J.A. 1990. A method which gives an objective measure of colonization of roots by vesicular–arbuscular mycorrhizal fungi. New Phytologist. 115: 495–501).
Point 5: Results presentation must be improved
Response 5: Results presentation was improved through i) a change of figures (format, histograms colour, organisation), ii) According to the next comment (Point 10), the authors reorganised the results into parts. The first one is “carbon pools” and the second one “soil microbial activity”. This organisation will improve the logical flow of the scientific messages of the present paper.
Point 6: Conclusion supported by results can be improved
Response 6: The conclusion was revised, amended and reorganised.
Point 7: The main effects of NT and crop rotation on the variables analyzed were not adequately revised
Response 7: The effects of all tillage treatments and crop rotation treatments were verified and revised in the results section and some paragraphs were amended and others were simplified. Figures 3 to 6 were improved (format and legend).
Point 8: Soil sampling is an important issue in this kind of study. This is the reason why soil layers collected should be stratified taking into account the treatments tested: thus, it is right to collect samples in tinny soil layers (e.g. 0-5 cm) of the NT plots, but it is not advisable to collect samples in stratified way in layers where the soil was mixed up to 30 cm.... Thus, it is advisable to show the dataset following the principles of soil sampling and the effects of treatments normally reported for NT and CT plots.
Response 8: Authors agree completely with the comment on the importance of soil sampling stratification mainly under NT. However, the 0-5 cm, where accumulation of organic residues is observed in NT, is included in the 0-20 cm. So, the organic carbon content measurement includes this tinny soil layer, allowing the comparison with the other tilled treatments. On the other hand, in the Mediterranean region and specifically in Tunisia the main issue of NT practice is the low level of residue restitution on soil surface. Indeed, in Tunisia the residues are considered an important feed resource for livestock in the summer period and farmers export the totality of residues kept on the soil after harvesting (Moujahed et al., 2015, https://doi.org/10.5897/AJAR2015.10396). Authors mentioned this information in the discussion section (L297-L299). Thereby, the soil is slightly covered both under CT and NT and the first 20 cm are relatively homogeneous.
Point 9: The influence of crop rotation on soil C pools and soil biota activity is related with the amount and composition of biomass added to soil in each agricultural season. Thus, to have a correct nexus of causality, it would be advisable to present in a table all the data related to crop yield or cover crop biomass production for each agricultural season.
Response 9: The SOCS is built gradually by the annual amount of crop residues. Before trial setup the plot was uncultivated for several years. With the different tested treatments, a part of the vegetal biomass returns to soil, which leads to an increase of the SOCS in all treatments (NT>MT>CT) (the result section was amended by these informations linked to the evolution of soil organic carbon stock compared to initial level).
The average biological yields data for the different treatments is below (additional data can be provided), explaining the increase of the SOCS from 2009 and showing a slight variation of biomass between treatments. On the Other hand, the quality of the successive returned residues (data not measured) had without doubt an influence on the SOCS.
|
Treatments |
Average biological yield (T/ha) |
|
CT-M |
4,36±4,30 |
|
CT-BI |
5,81±3,21 |
|
CT-TRI |
6,55±3,63 |
|
MT-M |
4,37±4,18 |
|
MT-BI |
6,05±3,61 |
|
MT-Tri |
6,73±3,83 |
|
NT-M |
5,23±5,13 |
|
NT-BI |
6,21±3,07 |
|
NT-TRI |
6,77±3,37 |
Point 10: Soil Health “the capacity of soil to function as a vital living system, within ecosystem and land-use boundaries, to sustain plant and animal productivity, maintain or enhance water and air quality, and promote plant and animal health” (Doran and Zeiss, 2000) is an important and ample concept that could not be used in this context. It was only evaluated 2 soil C pools and microbial activity and population of only a mycorrhiza group.
Response 10: Authors agree completely with the definition of the concept of soil health. The present study is mainly based on soil organic carbon and soil microbial aspects through: soil microbial respiration, soil microbial biomass carbon and arbuscular mycorrhizal fungal root colonization. Some authors asked that biological properties such as soil microorganisms were considered as an essential composition in soil health as well (recent review paper https://doi.org/10.1016/j.gecco.2020.e01118). A recent review papers of Yang et al (2020) (https://doi.org/10.1016/j.gecco.2020.e01118) and Karlen et al (2019) in USA (https://doi.org/10.1016/j.still.2019.104365) highlighted that the main indicators of soil health are organic carbon and soil microbial activity.
In the title, “soil health” is replaced by the different soil parameters followed in the present study such as soil organic carbon pools, global and specific soil microbial activity.
Point 11: Wheat as the main cash crop should be mentioned in the title of the manuscript, as well as the terms "C pools", "tillage-rotation interaction" and "microbial activity in a clayey Mediterranean soil.
Response 11: The revised title according to the proposition: Long term effects of tillage-crop rotation interaction on soil organic carbon pools and microbial activity of a wheat-based system in the Mediterranean region.
Round 2
Reviewer 1 Report
Dear Authors,
Thank you for taking my comments and suggestions into account and for comprehensive answers. Currently, the manuscript has been significantly improved; therefore, I recommend it for publication.
Reviewer 2 Report
Dear Authors. My arequests were attended. Thus, in my opinion, the paper could be accepted in its current form.